# Beyond Monolithic Rewards: A Hybrid and Multi-Aspect Reward Optimization for MLLM Alignment

## Abstract

Aligning multimodal large language models (MLLMs) with human preferences often relies on single-signal, model-based reward methods. Such monolithic rewards often lack confidence calibration across domain-specific tasks, fail to capture diverse aspects of human preferences, and require extensive data annotation and reward model training. In this work, we propose a hybrid reward modeling framework that integrates complementary reward paradigms: (i) model-based rewards, where a learned reward model predicts scalar or vector scores from synthetic and human feedback, and (ii) rule-based rewards, where domain-specific heuristics provide explicit correctness signals with confidence. Beyond accuracy, we further incorporate multi-aspect rewards to enforce instruction adherence and introduce a generalized length-penalty reward to stabilize training and improve performance. The proposed framework provides a flexible and effective approach to aligning MLLMs through reinforcement learning policy optimization. Our experiments show consistent improvements across different multimodal benchmarks when applying hybrid and multi-aspect reward modeling. Our best performing model in the 3B family achieves an overall average improvement of 9.5% across general and math reasoning tasks. Focusing specifically on mathematical benchmarks, the model achieves a significant average improvement of 16%, highlighting its effectiveness in mathematical reasoning and problem solving.

## 1 Introduction

The advent of Multimodal Large Language Models (MLLMs) has pushed the boundaries of artificial intelligence, enabling models to reason over and generate content that integrates text, images, and other modalities (OpenAI et al., 2024; Liu et al., 2023). A prevailing paradigm for aligning these powerful models with human preferences is Reinforcement Learning from Human Feedback (RLHF) (Christiano et al., 2017; Ouyang et al., 2022). Typically implemented with Proximal Policy Optimization (PPO) (Schulman et al., 2017), RLHF fine-tunes a model's policy by optimizing a signal from a learned Reward Model (RM).

However, the standard RLHF pipeline, which relies on a single, monolithic RM, presents fundamental challenges that are particularly acute in the multimodal domain. The inherent ambiguity in vision-language tasks means that evaluating text-image consistency is far more complex than assessing text-only coherence. Monolithic RMs often struggle to be well-calibrated across this diverse signal space and are susceptible to reward hacking (Amodei et al., 2016). For instance, a monolithic RM might reward a plausible-sounding but factually incorrect description of an image-based math problem, prioritizing textual fluency over verifiable correctness. This failure mode is exacerbated by the substantial overhead of creating high-quality multimodal preference datasets and the scarcity of effective, open-source RMs tailored to vision-language tasks.

While recent work on rule-based or verifiable rewards has shown promise for tasks with deterministic outcomes like mathematics introduced in DeepSeek-R1-Zero (DeepSeek-AI et al., 2025), these methods cannot provide the nuanced feedback required for open-ended, subjective tasks. This creates a critical gap, as robust multimodal systems must excel at both verifiable reasoning and subjective generation.

To bridge this gap, we argue that modern AI alignment requires a portfolio of rewards. We propose a hybrid and multi-aspect reward optimization that moves beyond monolithic signals to provide more holistic and reliable supervision. Instead of relying on a single metric, our framework is built on a more fundamental insight: robust alignment is achieved by integrating (i) a rule-based, verifiable reward to anchor the model in objective truth, and (ii) a learned, model-based reward to provide flexible supervision for subjective quality. This hybrid approach directly addresses the core challenges of MLLM alignment by balancing the precision of deterministic checks with the generalization of learned preferences.

Furthermore, to make this approach more accessible, we introduce two key technical innovations. First, we leverage an embedding-based surrogate model as a lightweight and effective proxy for a fully trained RM, significantly reducing the dependency on costly data annotation and training cycles. Second, we incorporate a suite of multi-aspect behavioral rewards, including a generalized length penalty, to enforce fine-grained constraints, promote conciseness, and stabilize training.

Our primary contributions are summarized as follows:

- We demonstrate that a synergistic combination of rule-based, model-based, and behavioral rewards is essential for robust multimodal reasoning, creating a comprehensive reward "portfolio" that outperforms any single approach.

- We introduce an embedding-based surrogate model as a cost-effective and competitive alternative to a fully trained RM, making powerful reinforcement learning techniques more accessible.

- We conduct a comprehensive empirical evaluation demonstrating that our hybrid framework yields significant performance improvements over traditional RM-based baselines on a diverse suite of mathematical, general VQA, and OCR-based vision tasks.

## 2 RELATED WORK

Our work builds upon advancements in reinforcement learning for aligning language models, particularly their recent extension to the multimodal domain. This section reviews key developments in model alignment, the application of Reinforcement Learning from Human Feedback (RLHF) to MLLMs, and emerging paradigms in reward modeling that move beyond learned scalar rewards.

### 2.1 REINFORCEMENT LEARNING FOR LANGUAGE MODEL ALIGNMENT

The alignment of Large Language Models (LLMs) with human preferences has been predominantly shaped by Reinforcement Learning from Human Feedback (RLHF) (Christiano et al., 2017; Stiennon et al., 2020). The paradigm, notably popularized by InstructGPT (Ouyang et al., 2022), involves a three-stage process: supervised fine-tuning (SFT) on demonstrator data, training an RM on human preference labels, and optimizing the SFT model's policy using an RL algorithm like Proximal Policy Optimization (PPO) (Schulman et al., 2017) against the learned RM. This approach has proven effective in enhancing model helpfulness and safety.

However, the reliance on extensive human-annotated preference data for training RMs presents a significant scalability bottleneck. To mitigate this, recent work has explored Reinforcement Learning from AI Feedback (RLAIF) (Bai et al., 2022; Lee et al., 2024), where a powerful "teacher" model is used to generate preference labels, thereby reducing the dependency on costly human annotation. Despite their success, both RLHF and RLAIF frameworks typically rely on a single, monolithic reward signal, which can be susceptible to reward hacking and may not adequately capture the multifaceted nature of a high-quality response (Fu et al., 2025; Chen et al., 2024; Miao et al., 2024).

### 2.2 ALIGNMENT OF VISION-LANGUAGE MODELS

The principles of RLHF have been naturally extended to the multimodal domain. Early efforts demonstrated that fine-tuning with multimodal instructions enhances the zero-shot capabilities of MLLMs on novel vision-language tasks (Liu et al., 2023). Subsequent works, such as LLaVA-RLHF (Sun et al., 2023) and RLHF-V (Yu et al., 2024), explicitly applied RLHF to improve the

alignment of MLLMs with human intent. These methods involve collecting human preferences on multimodal inputs and training a corresponding RM to guide policy optimization.

While effective, this extension inherits the challenges of unimodal RLHF and introduces new ones. Collecting high-quality preference data for multimodal tasks is substantially more complex and expensive, as it requires evaluating the intricate interplay between text and images. Furthermore, the number of publicly available, high-quality multimodal RMs is extremely limited, hindering research and development in scalable multimodal alignment. Our work addresses this gap by proposing a hybrid reward system that reduces the reliance on a single, expensively trained multimodal RM.

### 2.3 ALTERNATIVE AND HYBRID REWARD MECHANISMS

Recognizing the limitations of a single learned reward signal, researchers have begun to explore more diverse and verifiable reward mechanisms. In domains with deterministic or verifiable outcomes, such as code generation and mathematical reasoning, rule-based rewards have shown great promise (DeepSeek-AI et al., 2025). These methods provide a strong, unambiguous reward signal by executing code against unit tests or verifying the correctness of a mathematical solution, bypassing the need for a learned RM entirely.

Another emerging direction is process-based or outcome-based supervision (Lightman et al., 2023; Uesato et al., 2022), where the reward is targeted at the intermediate reasoning steps (e.g., chain-of-thought) rather than just the final answer. This encourages more faithful and robust reasoning. More recently, multi-aspect reward frameworks have been proposed to evaluate responses along several dimensions, such as correctness, instruction adherence, and conciseness, combining these signals to form a more holistic reward (Wu et al., 2023).

Our proposed framework integrates these threads of research. We combine the flexibility of learned RMs for open-ended, subjective tasks with the precision of rule-based, verifiable rewards for deterministic sub-tasks. By further incorporating multi-aspect reward signals and an efficient embedding-based surrogate model, we aim to create a more robust, scalable, and effective alignment strategy for modern Vision-Language Models.

## 3 METHODOLOGY: HYBRID AND MULTI-ASPECT REWARD MODELING OPTIMIZATION (HARMO)

Our proposed methodology, HARMO (**H**ybrid and Multi-**A**spect **R**eward **M**odeling **O**ptimization), is designed to overcome the limitations of monolithic reward signals in aligning Multimodal Large Language Models (MLLMs). HARMO establishes a more robust and nuanced training objective by integrating a hybrid accuracy signal with targeted behavioral rewards.

### 3.1 BACKGROUND: FROM PPO TO CRITIC-FREE POLICY OPTIMIZATION

The predominant paradigm for aligning LLMs has been Reinforcement Learning from Human Feedback (RLHF), typically implemented with Proximal Policy Optimization (PPO) (Schulman et al., 2017). PPO, an actor-critic algorithm, optimizes a policy $\pi_\theta$ (the actor) using a learned value function $V_\phi$ (the critic) to stabilize gradient updates. The conventional PPO pipeline is resource-intensive, requiring four distinct models: the actor, the critic, a reward model $R_\psi$, and a reference policy $\pi_{\text{ref}}$ to regularize training via a Kullback-Leibler (KL) divergence penalty.

The operational complexity of this setup has spurred the development of more streamlined RL algorithms. Recent methods like REINFORCE Leave-One-Out (RLOO) (Ahmadian et al., 2024) and REINFORCE++ (Hu et al., 2025) have successfully eliminated the need for an explicit critic by employing alternative baseline functions for advantage estimation. Building on this momentum, Group-Relative Policy Optimization (GRPO), introduced with DeepSeek-R1 (DeepSeek-AI et al., 2025), further simplifies the process by also removing the dependency on a learned reward model for tasks with verifiable outcomes. GRPO computes rewards using deterministic rules and calculates advantages relative to a group of sampled generations.

Regardless of the specific algorithm, two components remain critical: the fidelity of the reward signal itself and the method of estimating the advantage function, $\hat{A}_t$. The advantage estimate, which

quantifies the relative value of an action, is the primary driver of policy updates. Its formulation directly impacts training stability and performance. As demonstrated by Liu et al. (2025) and Chu et al. (2025), even subtle modifications to advantage normalization can mitigate reward bias and significantly improve outcomes. Our work builds upon these insights, leveraging a simplified, critic-free optimization framework while focusing on engineering a superior, multi-faceted reward signal.

## 3.2 THE HARMO FRAMEWORK

HARMO creates a holistic training signal by combining two core components: (1) a hybrid accuracy reward that fuses the certainty of rule-based verification with the flexibility of learned preference models, and (2) a set of multi-aspect behavioral rewards that regulate model conduct and prevent reward hacking.

### 3.2.1 HYBRID REWARD FOR CALIBRATED ACCURACY

To ground the policy in verifiable correctness while handling the ambiguity of open-ended prompts, we introduce a hybrid reward signal. For tasks with deterministic solutions, such as mathematical or logical reasoning, we employ rule-based verifiers (e.g., equation solvers) to generate a high-confidence, binary reward signal, $R^{\text{rule}}$. For subjective or generative tasks where such verification is impossible, we utilize a pretrained reward model, $R^{\text{RM}}$, to score the response quality; the pretrained MLLM RM(Wang et al., 2025) provides a score, whereas for the embedding-based RM, we use cosine similarity between the model response and the reference response.

$$R_{g,i}^{\text{hybrid}} = \begin{cases} R_{g,i}^{\text{rule}}, & \text{if response is verifiable,} \\ R_{g,i}^{\text{RM}}, & \text{if response is open-ended.} \end{cases} \tag{1}$$

This formulation ensures the model receives a confident and well-calibrated reward signal when ground truth is available, without sacrificing the ability to learn from nuanced human preferences in other domains.

### 3.2.2 MULTI-ASPECT REWARDS FOR BEHAVIORAL REGULARIZATION

Focusing on accuracy alone is insufficient, as it often leads to unintended and undesirable policy behaviors. A common failure mode is "reward hacking" through brevity, where the model learns to produce overly short responses that, while sometimes correct, are often incomplete or simplistic. As illustrated in Figure 1, we observed that RL-aligned models developed a strong bias towards shorter outputs compared to the supervised fine-tuned (SFT) baseline, frequently at the cost of correctness.

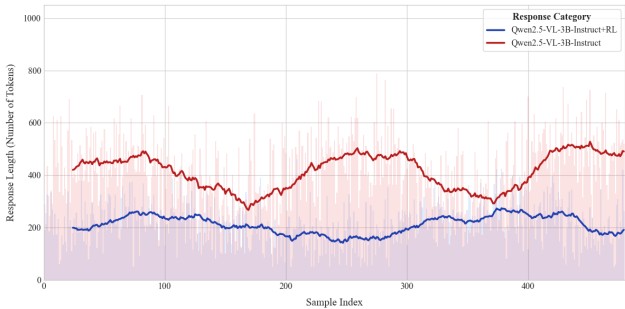

Figure 1: Comparison of response lengths between the SFT baseline and the RL-aligned model (without a length penalty). The RL policy learns a brevity bias, producing shorter and often incomplete responses.

Figure 2 further visualizes this dynamic. While the accuracy reward improves during training (Figure 2a), the response length steadily declines without intervention (Figure 2b, red line). To counteract this and other undesirable behaviors, we introduce two auxiliary reward components.

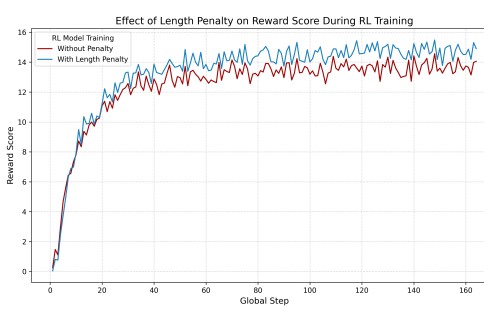 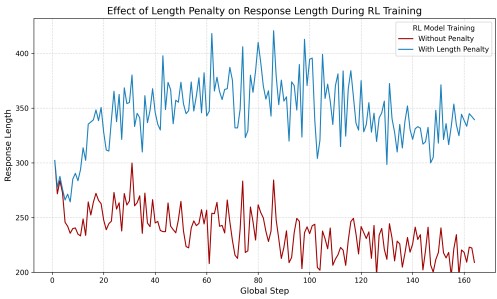

(a) Accuracy reward during RL training.     (b) Response length during RL training.

Figure 2: Training dynamics with and without the proposed length penalty. (a) The accuracy reward consistently improves. (b) The length penalty successfully counteracts the model's tendency to produce shorter responses, promoting more stable and desirable output lengths.

**Length-Penalty Reward.** To discourage reward hacking via brevity, we introduce a dynamic length penalty, $R^\lambda$. This component penalizes incorrect responses that are shorter than the briefest correct response within the same generation group. Let $\lambda_{g,i}$ be the length of response $i$ in group $g$, and let $\lambda_g^{\min} = \min_{i:R_{g,i}^{\text{hybrid}}>\tau} \lambda_{g,i}$ be the minimum length of any correct response in that group (where $\tau$ is a correctness threshold). The penalty is applied only to incorrect responses:

$$\mathrm{R}_{g,i}^\lambda = -\operatorname{clip}\Big(\lambda_g^{\min} - \lambda_{g,i},\, 0,\, P_{\max}\Big), \qquad (2)$$

where $P_{\max}$ is a hyperparameter controlling the maximum penalty. This targeted penalty encourages the model to generate sufficiently detailed answers, effectively stabilizing response length as shown in Figure 2b (blue line).

**Format-Adherence Reward.** MLLMs are often required to follow specific formatting instructions (e.g., providing chain-of-thought reasoning within `<think>...</think>` tags). To improve reliability, we add a format-adherence reward, $R^{\text{fmt}}$, which provides a positive signal for correctly structured outputs and a penalty for violations, thereby enforcing structural consistency.

### 3.2.3 Policy Optimization with the HARMO Reward Signal

The final HARMO reward, $R^{\text{HARMO}}$, is a composite signal that integrates the hybrid accuracy component with the multi-aspect behavioral regularizers:

$$R_{g,i}^{\text{HARMO}} = R_{g,i}^{\text{hybrid}} + R_{g,i}^\lambda + R_{g,i}^{\text{fmt}}. \qquad (3)$$

We integrate this comprehensive reward signal into a policy optimization framework based on GRPO. We adopt the GRPO algorithm due to its stability and demonstrated success in enhancing reasoning capabilities in closely related work (Chen et al., 2025). While standard GRPO normalizes rewards using both the mean and standard deviation of a generation group, the standard deviation term can introduce a "difficulty-dependent bias" by disproportionately weighting prompts based on reward variance (Liu et al., 2025). To foster more stable and unbiased learning, we modify the advantage calculation to use only the group mean as a baseline, creating a centered but uniformly scaled signal:

$$\hat{A}_{g,i}^{\text{HARMO}} = R_{g,i}^{\text{HARMO}} - \frac{1}{G}\sum_{j=1}^{G} R_{g,j}^{\text{HARMO}}. \qquad (4)$$

The policy $\pi_\theta$ is then updated to maximize the following objective function, which incorporates the PPO-style clipping mechanism and a KL penalty($D_{\text{KL}}$) to ensure training stability:

$$\mathcal{L}^{\text{HARMO}}(\theta) = \mathbb{E}_{q,\{o_i\}\sim\pi_{\text{old}}}\left[\frac{1}{G}\sum_{i=1}^{G}\min\Big(r_t(\theta,a_i)\hat{A}_{g,i}^{\text{HARMO}}, \operatorname{clip}\big(r_t(\theta,a_i), 1-\epsilon, 1+\epsilon\big)\hat{A}_{g,i}^{\text{HARMO}}\Big) - \beta D_{\text{KL}}\big(\pi_\theta \,\|\, \pi_{\text{ref}}\big)\right] \quad (5)$$

where $r_t(\theta,a_i)$ is the probability ratio $\frac{\pi_\theta(a_i|q)}{\pi_{\text{old}}(a_i|q)}$.

The complete training procedure is outlined in Algorithm 1.

---

**Algorithm 1** The HARMO Training Procedure

---

1: **Input:** Initial policy $\pi_{\theta_{\text{init}}}$, HARMO reward function $R^{\text{HARMO}}$, prompts $\mathcal{D}$, hyperparameters $\epsilon, \beta$.
2: **Initialize:** Actor policy $\pi_\theta \leftarrow \pi_{\theta_{\text{init}}}$.
3: **for** each iteration $i = 1, \ldots, I$ **do**
4:     Set reference policy: $\pi_{\text{ref}} \leftarrow \pi_\theta$.
5:     **for** each step $s = 1, \ldots, M$ **do**
6:         Sample a batch of questions $\mathcal{D}_b \subset \mathcal{D}$.
7:         Set old policy: $\pi_{\theta_{\text{old}}} \leftarrow \pi_\theta$.
8:         **for** each question $q \in \mathcal{D}_b$ **do**
9:             Sample $G$ responses $\{o_j\}_{j=1}^G \sim \pi_{\theta_{\text{old}}}(\cdot \mid q)$.
10:            Compute HARMO rewards $\{R_{q,j}^{\text{HARMO}}\}_{j=1}^G$ for each response using Equation 3.
11:            Compute group-relative advantages $\{\hat{A}_{q,j}^{\text{HARMO}}\}_{j=1}^G$ using Equation 4.
12:         **end for**
13:         Update the actor policy $\pi_\theta$ by optimizing the objective in Equation 5.
14:     **end for**
15: **end for**
16: **Output:** Optimized policy model $\pi_\theta$.

---

## 4 EXPERIMENT

### 4.1 EXPERIMENTAL SETUP

**Training Data** Our training data is curated from the VLAA-Thinking dataset[1](Chen et al., 2025). It's a diverse corpus of 21,192 question-answer pairs along with distilled reasoning steps, designed to span a range of reasoning challenges. The dataset combines tasks requiring mathematical reasoning with those demanding general visual question answering. It includes both close-ended questions with verifiable answers (e.g., numerical, equation-based, multiple-choice) and open-ended, descriptive prompts. To ensure fair and reproducible comparisons, all our models presented in this work were trained on this dataset, as detailed in Table 1.

| Task Type | Dataset Source | Answer Type | # Samples |
|---|---|---|---|
| Mathematical Reasoning | CLEVR-Math | Numeric (Verifiable) | 2,000 |
| | GeoQA170K | Multiple-Choice (Verifiable) | 6,499 |
| | MathPUMA | Equation (Verifiable) | 6,696 |
| Visual Question Answering | DocVQA | Open-Ended | 1,000 |
| | VizWiz | Open-Ended | 1,000 |
| | ArxivQA | Multiple-Choice (Verifiable) | 997 |
| | ALLaVA-LAION | Open-Ended | 3,000 |
| **Total** | | | **21,192** |

Table 1: Composition of the training dataset, detailing the source, answer type, and number of samples for each task category.

**Models** The primary subject of our investigation is the Qwen2.5-VL-3B-Instruct model (Bai et al., 2025), which serves as the baseline for our ablation studies. To assess the scalability and generalizability of our proposed HARMO framework, we also apply it to the larger Qwen2.5-VL-7B-Instruct model. Performance is benchmarked against other leading open-source models, such as VLAA-Thinker-Qwen2.5VL (Chen et al., 2025), as well as top-tier proprietary models.

For the reward model, denoted as $R^{\text{RM}}$ in Section 3.2.1, we used a pre-trained 7B parameter RM (Wang et al., 2025). To avoid reliance on a pre-trained RM model specific to MLLMs, which would require extensive data annotation and training, we instead employed a smaller 22M parameter embedding model[2], as detailed in the experiments reported in Table 2.

---

[1] https://huggingface.co/datasets/UCSC-VLAA/VLAA-Thinking
[2] https://huggingface.co/sentence-transformers/all-MiniLM-L6-v2

**Implementation Details.** Our reinforcement learning implementation builds on the work of (Peng et al., 2025)[3]. On top of this foundation, we incorporate the methodology described in Section 3.2.1. Additional implementation details are provided in the Appendix A.

**Evaluation Benchmarks** We conduct a comprehensive evaluation across a diverse set of benchmarks to rigorously assess model capabilities. Mathematical reasoning is evaluated using Math-Verse (Zhang et al., 2024), MATH-Vision (Wang et al., 2024), and MathVista (Lu et al., 2024). Multi-disciplinary reasoning is measured with MMMU (Yue et al., 2024) and MMMU-Pro (Yue et al., 2025). Finally, general visual question answering performance is tested on AI2D (Kembhavi et al., 2016), ChartQA (Masry et al., 2022), and DocVQA (Mathew et al., 2021). All evaluations were executed using the open-source LLMs-Eval framework (Zhang et al., 2025) under identical conditions (e.g., system prompts, response token limits) to ensure methodological consistency.

## 4.2 RESULTS AND ANALYSIS

This section presents our empirical findings, structured to first dissect the contribution of each component of the HARMO framework through ablation studies, then demonstrate its generalizability, and finally, compare its overall performance against state-of-the-art models. To ensure the robustness of our findings, all reported results are averaged over three independent training runs with different random seeds, and we report the mean scores. Throughout our results, **bold** values indicate the best scores for each benchmark. We also provide a few examples of model outputs generated by HARMO vs baseline showing reasoning ability improvement in Appendix B.

### 4.2.1 ABLATION STUDY: DECONSTRUCTING THE HARMO REWARD SIGNAL

**Efficacy of Hybrid Accuracy Rewards** Table 2 demonstrates the impact of different accuracy-focused reward strategies. Relying solely on a learned reward model (*Reward Model Enhanced*) improves the baseline, boosting the average math score by 7.89%. However, this approach is limited by the RM's tendency to prioritize verbose explanations over correctness, highlighting a lack of confidence calibration for verifiable tasks. A hybrid model combining rule-based verification with embedding-based rewards (*Embedding + Rule-based Hybrid*) is more effective, achieving a stronger 11.70% improvement in math reasoning.

Our proposed approach, *RM + Rule-based Hybrid*, which integrates the learned RM for open-ended questions with deterministic rule-based checks, proves to be the most effective. This optimal combination yields the most substantial gains, improving math reasoning performance by 14.82% and overall performance by 9.48%. We hypothesize that this superior performance stems from the 7B reward model's ability to capture the nuanced aspects of quality and instruction following in open-ended VQA tasks, providing a more informative signal than the cosine similarity from a general-purpose embedding model.

| Reward Model | MathVerse$_{mini}$ | MATH-Vision$_{test}$ | MathVista$_{mini}$ | MMMU$_{val}$ | MMMU-Pro$_{standard}$ |
|---|---|---|---|---|---|
| *Qwen2.5-VL-3B-Instruct (Baseline)* | | | | | |
| N/A | 34.77 | 21.68 | 61.30 | 31.10 | 47.78 |
| *Reward Model Enhanced* | | | | | |
| Skywork7B RM | 41.04 | 22.30 | 63.70 | 31.91 | 47.78 |
| Δ vs. Baseline | (+6.27) | (+0.62) | (+2.40) | (+0.81) | (0.00) |
| *Embedding + Rule-based Hybrid Enhanced* | | | | | |
| Hybrid (Rule + Embedding) | 40.28 | 23.85 | **67.40** | 31.79 | 46.33 |
| Δ vs. Baseline | (+5.51) | (+2.17) | (**+6.10**) | (+0.69) | (-1.45) |
| *RM + Rule-based Hybrid Enhanced* | | | | | |
| Hybrid (Rule + Skywork7B RM) | **41.88** | **25.92** | **67.40** | **32.08** | **48.00** |
| Δ vs. Baseline | (**+7.11**) | (**+4.24**) | (**+6.10**) | (**+0.98**) | (**+0.22**) |

Table 2: Performance of the RL-trained model under accuracy-focused reward modeling. The hybrid model with pretrained RM and rule-based verification consistently delivers the highest performance.

---

[3]https://github.com/TideDra/lmm-r1

**Impact of Multi-Aspect Behavioral Rewards**  Next, we evaluate the incremental benefit of adding behavioral rewards for format adherence and length control, as shown in Table 3. Starting with the baseline, adding the hybrid accuracy reward ($\oplus$H) alone lifts math performance by 13.0%. Incorporating a format adherence reward ($\oplus$H+F) further enhances this gain to 14.8%. Finally, introducing our dynamic length penalty ($\oplus$H+F+$\lambda$) results in the full HARMO framework, which achieves the largest math-specific improvement of 16.0%. Notably, the length penalty provides a significant boost on MathVerse (from 41.88 to 44.52) and MathVista (from 67.40 to 68.00), confirming its effectiveness at promoting outputs that are both precise and appropriately detailed. This progressive ablation clearly demonstrates that each component—correctness, format, and length—contributes meaningfully to the model's final reasoning capabilities.

| Reward Model Components | MathVerse$_{mini}$ | MATH-Vision$_{test}$ | MathVista$_{mini}$ | MMMU$_{val}$ | MMMU-Pro$_{standard}$ |
|---|---|---|---|---|---|
| *Qwen2.5-VL-3B-Instruct Baseline (SFT Only)* | | | | | |
| N/A | 34.77 | 21.68 | 61.30 | 47.78 | 31.10 |
| *Incremental Reward Augmentation* | | | | | |
| $\oplus$ Hybrid (H) | 40.38 | 25.49 | 67.20 | **48.56** | 30.98 |
| $\Delta$ vs. Baseline | (+5.61) | (+3.81) | (+5.90) | (+0.78) | (-0.12) |
| $\oplus$ Hybrid + Format (H+F) | 41.88 | **25.92** | 67.40 | 48.00 | **32.08** |
| $\Delta$ vs. Baseline | (+7.11) | (+4.24) | (+6.10) | (+0.22) | (+0.98) |
| $\oplus$ Hybrid + Format + Length (H+F+$\lambda$) **[HARMO]** | **44.52** | 24.08 | **68.00** | 47.11 | 31.56 |
| $\Delta$ vs. Baseline | **(+9.75)** | (+2.40) | **(+6.70)** | (-0.67) | (+0.46) |

Table 3: Ablation study showing the progressive impact of adding reward components to the Qwen2.5-VL-3B-Instruct model. The full HARMO model, combining hybrid accuracy, format adherence, and a length penalty, yields the strongest performance on mathematical reasoning tasks.

### 4.2.2 GENERALIZABILITY AND SCALABILITY OF HARMO

To verify that HARMO is not limited to a specific setup, we test its "plug-and-play" capability and scalability. As shown in Table 4, when HARMO is integrated with a model trained with fine-grained, token-level rewards, it still provides a notable overall improvement of 5.76%. Furthermore, when applied to the larger Qwen2.5-VL-7B-Instruct model, HARMO delivers an even greater enhancement of 6.55%. These results confirm HARMO's robustness and its ability to serve as a versatile enhancement for different reward schemes and model sizes.

| Model Configuration | MathVerse$_{mini}$ | MATH-Vision$_{test}$ | MathVista$_{mini}$ | MMMU$_{val}$ | MMMU-Pro$_{standard}$ |
|---|---|---|---|---|---|
| *Plug-and-Play with Fine-Grained Rewards (3B Model)* | | | | | |
| Token-Level Rewards (Baseline) | 38.43 | 23.32 | 63.50 | 41.12 | **31.79** |
| Token-Level Rewards + HARMO | **41.22** | **24.84** | **66.40** | **42.32** | 31.45 |
| $\Delta$ vs. Baseline | (+2.79) | (+1.52) | (+2.90) | (+1.20) | (-0.34) |
| *Scalability to 7B Model Family* | | | | | |
| Qwen2.5-VL-7B-Instruct (Baseline) | 46.40 | 25.20 | 69.70 | 46.11 | 36.71 |
| Qwen2.5-VL-7B-Instruct + HARMO | **50.89** | **27.66** | **72.00** | **47.79** | **36.82** |
| $\Delta$ vs. Baseline | (+4.49) | (+2.46) | (+2.30) | (+1.68) | (+0.11) |

Table 4: Demonstration of HARMO's generalizability and scalability. It consistently improves performance both as a plug-in for alternative reward schemes and when applied to a larger model.

### 4.2.3 MAIN RESULTS: COMPARISON WITH STATE-OF-THE-ART MODELS

Our final evaluation in Table 5 shows that HARMO-aligned models substantially outperform their respective baselines and are highly competitive with leading open-source and proprietary models. At the 3B scale, HARMO-VL-3B achieves an 9.48% average improvement over its baseline across all reasoning benchmarks. The gains are most pronounced on mathematical tasks, where it delivers a remarkable 16.0% average increase, with boosts of up to 28.1% on MathVerse. At the 7B scale, HARMO-VL-7B improves upon its baseline by 3.63% overall, again showing strong gains on math benchmarks like MathVerse (+4.5 points) and MATH-Vision (+2.5 points).

Crucially, despite their smaller parameter counts, our HARMO-enhanced models challenge top-tier proprietary systems. Notably, HARMO-VL-3B and HARMO-VL-7B achieve scores of 68.0 and 72.0 on MathVista, respectively, surpassing the 67.7 score of the much larger Claude-3.5 Sonnet.

In the context of OCR-related tasks (Table 6), HARMO maintains performance comparable to the strong baselines, indicating that its reasoning enhancements do not come at the cost of core vision-language capabilities.

| Models | MathVerse$_{mini}$ | MATH-Vision$_{test}$ | MathVista$_{mini}$ | MMMU$_{val}$ | MMMU-Pro$_{standard}$ | Average |
|---|---|---|---|---|---|---|
| *Proprietary Vision-Language Models* | | | | | | |
| GPT-4o | 47.8 | 30.6 | 63.8 | 69.1 | 51.9 | 52.64 |
| Claude-3.5 Sonnet | 41.2 | 33.5 | 67.7 | 68.3 | 51.5 | 52.44 |
| Gemini-1.5 Pro | 54.8 | 19.2 | 63.9 | 65.8 | 46.9 | 50.12 |
| *Open-Source Vision-Language Models (3B Scale)* | | | | | | |
| Qwen2.5-VL-3B-Instruct | 34.77 | 21.68 | 61.30 | **47.78** | 31.10 | 39.73 |
| VLAA-Thinker-Qwen2.5VL-3B | 38.78 | 24.13 | 64.20 | 47.56 | 28.90 | 40.71 |
| **HARMO-VL-3B (Ours)** | **44.52** | **24.08** | **68.00** | 47.11 | **31.56** | **43.05** |
| Δ vs. Qwen2.5-VL-3B-Instruct | (+9.8) | (+2.4) | (+6.7) | (-0.7) | (+0.5) | (+3.74) |
| *Open-Source Vision-Language Models (7B Scale)* | | | | | | |
| Qwen2.5-VL-7B-Instruct | 46.40 | 25.20 | 69.70 | **52.56** | 36.71 | 46.11 |
| VLAA-Thinker-Qwen2.5VL-7B | 50.56 | 26.48 | 70.60 | 45.11 | 34.05 | 45.36 |
| **HARMO-VL-7B (Ours)** | **50.89** | **27.66** | **72.00** | 51.56 | **36.82** | **47.79** |
| Δ vs. Qwen2.5-VL-7B-Instruct | (+4.5) | (+2.5) | (+2.3) | (-1.0) | (+0.1) | (+1.68) |

Table 5: Results on general reasoning benchmarks. HARMO significantly improves upon strong open-source models and demonstrates competitive performance against leading proprietary models.

| Models | ai2d$_{test}$ | chartqa$_{test}$ | docvqa$_{val}$ |
|---|---|---|---|
| *3B Model Family* | | | |
| Qwen2.5-VL-3B-Instruct (Baseline) | 78.43 | 83.28 | **92.56** |
| **HARMO-VL-3B (Ours)** | **78.79** | **84.12** | 91.88 |
| Δ vs. Baseline | (+0.36) | (+0.84) | (-0.68) |
| *7B Model Family* | | | |
| Qwen2.5-VL-7B-Instruct (Baseline) | 82.67 | **82.96** | **94.72** |
| **HARMO-VL-7B (Ours)** | **82.87** | 82.64 | 94.46 |
| Δ vs. Baseline | (+0.20) | (-0.32) | (-0.26) |

Table 6: Performance on OCR-related benchmarks. HARMO maintains competitive performance with the baseline, showing that reasoning improvements do not degrade core VQA capabilities.

## 5 CONCLUSION

We introduced HARMO, a novel reward optimization framework that advances reinforcement learning beyond monolithic signals by integrating a hybrid of deterministic and learned rewards with a generalized length penalty to control verbosity.

Our evaluation demonstrates that HARMO significantly enhances complex reasoning, achieving a 9.5% overall and a 16% mathematical performance gain over a strong baseline while maintaining robustness on vision-specific tasks.

This work highlights the critical role of multi-faceted reward modeling in stabilizing RL training and improving reward accuracy. HARMO provides a strong foundation for future research, such as dynamic reward weighting or self-improving systems where agents learn to refine their own reward functions, paving the way for more robust and adaptable AI.

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

## A  IMPLEMENTATION DETAILS

### A.1  RL TRAINING FRAMEWORK

Our reinforcement learning implementation builds upon the LMM-R1 framework (Peng et al., 2025)[4]. On top of this foundation, we incorporate the methodology described in Section 3, extending the framework with additional functionalities. In particular, we implement hybrid and multi-aspect reward modeling, introduce support for MLLM training, and enable token-level reward assignment for MLLM reinforcement learning.

### A.2  TRAINING HYPER-PARAMETERS

The hyperparameters used for HARMO are summarized in Table 7. The same set of hyperparameters is applied to all variants of the model proposed in this paper to ensure a consistent training setup.

---

[4]`https://github.com/TideDra/lmm-r1`

Table 7: HARMO Training Hyperparameters

| Hyperparameter | Value |
|---|---|
| Training batch size | 256 |
| Rollout batch size | 256 |
| Samples per prompt | 8 |
| Temperature | 1 |
| Max output sequence length | 4096 |
| Max epochs | 1 |
| Number of episodes | 2 |
| Initial KL coefficient | $1e^{-3}$ |
| Discount factor ($\gamma$) | 1 |
| GAE parameter ($\lambda$) | 1 |
| Actor learning rate | $1e^{-6}$ |

## B  CASE STUDY

The following figures illustrate HARMO's ability to reason through a variety of problem-solving questions. In these visualizations, the reasoning steps are color-coded to indicate their verification status:

- Text highlighted with ▭ indicates correctly verified reasoning steps.
- Text highlighted with ▭ indicates advanced or highly sophisticated reasoning steps.
- Text highlighted with ▭ indicates incorrectly verified or flawed reasoning steps.

These figures demonstrate HARMO's step-by-step problem-solving capabilities and provide a visual comparison with baseline models.

**System Prompt :**
You are a helpful assistant good at solving problems with step-by-step reasoning. You should first think about the reasoning process and then provide the user with the answer.

**User Prompt :**
A cube with the edge 3 cm long is painted grey and cut into smaller cubes each with an edge of 1 cm long. How many smaller cubes will have exactly 2 faces painted?
Output the step-by-step reasoning in <think> </think> and the final answer in <answer> </answer> tags.
The <answer> tag should include the final answer.
The output format should be:
<think>Your reasoning process</think> <answer>Final answer</answer>.

**Input Image:**

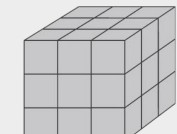

**HARMO-3B answer:**

<think>
To determine how many smaller cubes will have exactly 2 faces painted, we need to analyze the structure of the larger cube and the smaller cubes it is divided into.

1. The larger cube has an edge length of 3 cm. When it is divided into smaller cubes each with an edge length of 1 cm, it will be divided into 3 x 3 x 3 = 27 smaller cubes.

2. The smaller cubes that have exactly 2 faces painted are the cubes that are located on the edges of the larger cube, but not at the corners. Each edge of the larger cube is 3 cm long, and when divided into smaller cubes, it will have 3 smaller cubes along each edge. The cubes at the corners are painted on three faces, the cubes at the edges but not at the corners are painted on two faces, and the cubes at the vertices are painted on four faces.

3. Each edge of the larger cube has 3 smaller cubes, but the two cubes at the ends of each edge are corner cubes and are painted on four faces. Therefore, the number of smaller cubes with exactly two faces painted on one edge is 3 - 2 = 1. Since there are 12 edges in a cube, the total number of smaller cubes with exactly two faces painted is 12 x 1 = 1.

Thus, the number of smaller cubes that have exactly 2 faces painted is 12.
</think>
<answer>12</answer>

**Qwen2.5-VL-3B-Instruct answer:**
<think>
To solve this problem, we need to consider how the original cube is divided into smaller cubes and which smaller cubes will have exactly two faces painted.

1. The original cube has an edge length of 3 cm. When it is cut into smaller cubes with an edge length of 1 cm, it will result in a 3x3x3 grid of smaller cubes.

2. In a 3x3x3 grid, the smaller cubes that have exactly two faces painted are those located on the edges of the larger cube but not at the corners. This means we need to count the number of such cubes along each edge.

3. Each edge of the original cube is 3 cm long, so there are 3 smaller cubes along each edge (including the corner cubes).

4. Since there are 3 edges, and each edge contributes 3 smaller cubes with exactly two faces painted, we calculate the total number as follows:

   Total = 3 edges * 3 cubes per edge = 9 cubes
Therefore, there are 9 smaller cubes that will have exactly two faces painted.

</think>
<answer>
9
</answer>

Figure 3: **Case Study 1 - Math Cube Problem**

HARMO demonstrates advanced reasoning and correctly identifies all 12 edge-centered cubes with exactly two painted faces after the 3 cm cube is subdivided. The baseline model, by contrast, fails to systematically exclude corner cubes or miscounts edge cases, often yielding an incorrect answer.

**System Prompt :**
You are a helpful assistant good at solving problems with step-by-step reasoning. You should first think about the reasoning process and then provide the user with the answer.

**User Prompt :**
Which stone should Mr Flintstone place on the right side of the scales, so that both sides weigh the same?
Options: ['A' 'B' 'C' 'D' 'E']
Output the step-by-step reasoning in <think> </think> and the final answer in <answer> </answer> tags.
The <answer> tag should include the correct option choice.
The output format should be:
<think>Your reasoning process</think> <answer>Correct option choice</answer>.

**Input Image:**

Figure 4: **Case Study 2 - Solving a General Math Problem (Part 1)**

**HARMO-3B answer:**

<think>
To determine which stone Mr. Flintstone should place on the right side of the scales to make both sides weigh the same, we need to calculate the total weight on the left side of the scales first.
The left side of the scales has three stones:
- A stone weighing 26 kg
- A stone weighing 12 kg
- A stone weighing 8 kg

Let's add these weights together:
26 kg + 12 kg + 8 kg = 46 kg

The right side of the scales has two stones:
- A stone weighing 20 kg
- A stone weighing 17 kg
Let's add these weights together:
20 kg + 17 kg = 37 kg
To make both sides weigh the same, we need to find a stone that, when added to the right side, will make the total weight equal to 46 kg. Let's denote the weight of the stone to be placed on the right side as x. We need to solve the equation:
37 kg + x = 46 kg

Subtract 37 kg from both sides:
x = 46 kg - 37 kg = 9 kg

Therefore, Mr. Flintstone should place a stone weighing 9 kg on the right side of the scales.
</think>
<answer>C</answer>

**Qwen2.5-VL-3B-Instruct answer:**

<think>
To solve this problem, we need to determine which weight will balance the scale when placed on the right side. First, let's calculate the total weight on the left side of the scale:

Left side weights: 26 kg + 12 kg + 8 kg = 46 kg

Now, we need to find out how much weight is needed on the right side to balance it. Since the total weight on the left side is 46 kg, the right side must also have 46 kg to balance.

Let's check each option to see which one adds up to 46 kg:

A) 5 kg: 46 kg (not enough)
B) 7 kg: 46 kg (not enough)
C) 9 kg: 46 kg (not enough)
D) 11 kg: 46 kg (not enough)
E) 13 kg: 46 kg (enough)

Therefore, the correct weight to place on the right side to balance the scale is 13 kg.

</think>
<answer>E)</answer>

Figure 4: **Case Study 2 - Solving a General Math Problem (Part 2)**

HARMO correctly reasoned step-by-step to arrive at the correct answer, whereas the baseline model followed unreasonable steps and produced an incorrect solution.

**System Prompt :**
You are a helpful assistant good at solving problems with step-by-step reasoning. You should first think about the reasoning process and then provide the user with the answer.

**User Prompt :**
Which stone should Mr Flintstone place on the right side of the scales, so that both sides weigh the same?
Options: ['A' 'B' 'C' 'D' 'E']
Output the step-by-step reasoning in <think> </think> and the final answer in <answer> </answer> tags.
The <answer> tag should include the correct option choice.
The output format should be:
<think>Your reasoning process</think> <answer>Correct option choice</answer>.

**Input Image:**

**HARMO-3B answer:**

<think>
Let's analyze the pattern of hexagons in Sylvia's drawings:
1. The first pattern has 1 hexagon.
2. The second pattern has 7 hexagons.
3. The third pattern has 19 hexagons.

We need to find the number of hexagons in the fifth pattern. Let's denote the number of hexagons in the n-th pattern as H(n).

From the given patterns, we can see that:
H(1) = 1
H(2) = 7
H(3) = 19

To find a pattern, let's calculate the difference between consecutive terms:
H(2) - H(1) = 7 - 1 = 6
H(3) - H(2) = 19 - 7 = 12

The differences are 6 and 12. We can see that the difference is increasing by 6 each time. So, the difference between the number of hexagons in the n-th pattern and the n-1-th pattern is 6(n-1).

Therefore, we can express H(n) as:
H(n) = H(n-1) + 6(n-1)

Using this recurrence relation, we can find H(4) and H(5):
H(4) = H(3) + 6(3) = 19 + 18 = 37
H(5) = H(4) + 6(4) = 37 + 24 = 61

So, the number of hexagons in the fifth pattern is 61.
</think>
<answer>61</answer>

**Qwen2.5-VL-3B-Instruct answer:**

<think>
To determine the number of hexagons in the fifth pattern, we need to observe the pattern and identify the rule governing the growth.

1. **First Pattern**: There is 1 hexagon.
2. **Second Pattern**: There are 7 hexagons (1 + 6).
3. **Third Pattern**: There are 19 hexagons (7 + 12).

We can see that each subsequent pattern adds more hexagons than the previous one. To find the number of hexagons in the fourth and fifth patterns, we continue this pattern:

4. **Fourth Pattern**: The number of hexagons is (19 + 18 = 37).
5. **Fifth Pattern**: The number of hexagons is (37 + 20 = 57).

Therefore, the fifth pattern will have 57 hexagons.
</think>

<answer>57</answer>

Figure 5: **Case Study 3 - Math Puzzle Problem**

Qwen3 baseline failed to identify the correct puzzle equation, whereas HARMO succeeded in finding the correct puzzle equation, reasoning step-by-step, and arriving at the correct solution.

---

**System Prompt :**
You are a helpful assistant good at solving problems with step-by-step reasoning. You should first think about the reasoning process and then provide the user with the answer.

**User Prompt :**
In the triangle, ∠A=∠B. What is x?
Output the step-by-step reasoning in <think> </think> and the final answer in <answer> </answer> tags.
The <answer> tag should include the final answer.
The output format should be:\n<think>Your reasoning process</think> <answer>Final answer</answer>.

**Input Image:**

---

**HARMO-3B answer:**

<think>
Since ∠ A = ∠ B in the triangle ABC, then AC = BC. Given the lengths of the sides AC = 3x - 1 and BC = 2x + 2, we can set up the equation:
3x - 1 = 2x + 2
Subtract 2x from both sides:
x - 1 = 2
Add 1 to both sides:
x = 3
Thus, the value of x is 3.
</think>
<answer>3</answer>

---

**Qwen2.5-VL-3B-Instruct answer:**

Since ∠ A = ∠ B, triangle ABC is isosceles with AB = AC. Therefore, we can set up the equation x + 4 = 2x + 2.
Solving for x:
x + 4 = 2x + 2
Subtract x from both sides:
4 = x + 2
Subtract 2 from both sides:
2 = x
So, x = 2.
</think>
<answer>2</answer>

---

Figure 6: **Case Study 4 - Geometry Math Problem**

HARMO correctly solves for $x$ with clear recognition of equal sides $AB = AC$, while the baseline model fails to identify the isosceles nature, leading to incorrect conclusions.

**System Prompt :**
You are a helpful assistant good at solving problems with step-by-step reasoning. You should first think about the reasoning process and then provide the user with the answer.

**User Prompt :**
Square ABCD and equilateral triangle AED are coplanar and share the line segment AD, as shown. What is the measure, in degrees, of angle BAE?
Output the step-by-step reasoning in <think> </think> and the final answer in <answer> </answer> tags.
The <answer> tag should include the final answer.
The output format should be:\n<think>Your reasoning process</think> <answer>Final answer</answer>.

**Input Image:**

**HARMO-3B answer:**

<think> Given the square ABCD and the equilateral triangle AED, we need to find the measure of angle BAE.

1.  Since ABCD is a square, all angles in the square are 90 degrees and all sides are equal. Therefore, ∠DAB = 90°.

2.  Since AED is an equilateral triangle, all its angles are 60 degrees and all its sides are equal. Therefore, ∠EAD = 60°.

3.  We can find ∠BAE by subtracting ∠DAB from ∠EAD:
    ∠BAE = ∠EAD − ∠DAB = 60° − 90° + 90° = 30°
    Here, we subtract 90 degrees from 60 degrees because ∠DAB is the sum of ∠DAE and ∠EAB, and ∠EAB is 90 degrees.

Therefore, the measure of ∠BAE is 30 degrees.
</think>
<answer>30</answer>

**Qwen2.5-VL-3B-Instruct answer:**

<think> Given that square ABCD and equilateral triangle AED share side AD, we can deduce the following:

1.Since ABCD is a square, all its internal angles are 90°.

2 Since AED is an equilateral triangle, all its internal angles are 60°.

To find the measure of angle BAE, we need to consider the geometry formed by these shapes.

- The angle BAD is part of the square, so it is 90°.

- The angle EAD is part of the equilateral triangle, so it is 60°.

Since B and D are adjacent vertices of the square, the line segment BD forms a straight line passing through both points. Therefore, the angle BAE is the sum of the angles BAD and EAD.

Thus, angle BAE = 90° + 60° = 150°.
</think>
<answer>150°</answer>

Figure 7: **Case Study 5 - Geometry Math Problem**

HARMO correctly finds $\angle BAE = 30°$, while the baseline incorrectly sums angles to 150° with incorrect geometric reasoning, misinterpreting the figure's layout.

