# OpenReview forum: "Beyond Monolithic Rewards: A Hybrid and Multi-Aspect Reward Optimization for MLLM Alignment"
_ICLR.cc/2026/Conference — ICLR 2026 Conference Withdrawn Submission_

### Official Review · Reviewer_jZTn · 2025-10-23

**Soundness:** 1
**Presentation:** 2
**Contribution:** 1
**Rating:** 2
**Confidence:** 4

**Summary:**

This paper proposes HARMO, a hybrid and multi-aspect reward optimization framework for GRPO-style training. The core idea is a reward engineering technique that uses hybrid rewards, including rule-based reward, reward model-based reward, as well as multi-aspect rewards, including format reward, and lenght-penalty reward. HARMO outperforms baslines on various benchmarks.

**Strengths:**

1. This paper presents a clear and practical hybrid reward design: the combination of verifiable rule-based and open-ended RRM reward, which addresses a known weakness of monolithic scalar rewards in multimodal settings.
2. This paper proposes a GRPO variant with modified advantage: modifying the advantage calculation to use only the group mean as a baseline, which is a concrete algorithmic choice to reduce difficulty-dependent bias.

**Weaknesses:**

1. The paper is limited in novelty. The core algorithm is a modest variant of GRPO: the primary novelty lies in reward engineering, which is a simple combination of rule-based and reward model-based rewards. The format reward is standard; the length penalty is clever but relies on group sampling and the existence of at least one “correct” sample in the group.
2. Eq. 3 implies equal weighting, but the component scales differ. Without explicit weights or normalization, training may be dominated by whichever term has larger magnitude. There is no sensitivity analysis of P_{max}, τ, or R{fmt} magnitudes.
3. Advantage normalization change: Removing std is motivated by difficulty-dependent bias, but there is no ablation on this specific choice within HARMO.
4. The baselines in the experiments are not clearly expressed.
5. Open-ended scoring with cosine similarity to a single reference contradicts the claim that these tasks are not verifiable. It risks punishing diverse, equally good answers, especially in VQA with many acceptable phrasings.

**Questions:**

1. How are R^{hybrid}, R^λ, and R^{fmt} scaled or weighted relative to each other?
2. How to differentiate which responses are verifiable and which are open-ended?

---

> ### Author Response · Authors · 2025-11-29
>
> We would like to thank the reviewer for their helpful comments. We address the raised weaknesses and questions below.
>
> > [W1] The paper is limited in novelty. The core algorithm is a modest variant of GRPO: the primary novelty lies in reward engineering, which is a simple combination of rule-based and reward model-based rewards. The format reward is standard; the length penalty is clever but relies on group sampling and the existence of at least one “correct” sample in the group.
>
> HARMO’s novelty lies in systematically combining multiple reward types into a hybrid, multi-aspect framework for MLLM alignment. Unlike prior work (GRPO, DAPO) that focuses on deterministic or single-aspect rewards, HARMO unifies rule-based and open-ended rewards and incorporates length, format, and process-based feedback. Ablation studies (Tables 2–3) show that performance gains stem from this specific integration, not just the use of hybrid rewards. In short, HARMO enables robust supervision across both verifiable and open-ended multimodal tasks.
>
> > [W2] Eq. 3 implies equal weighting, but the component scales differ. Without explicit weights or normalization, training may be dominated by whichever term has a larger magnitude. There is no sensitivity analysis of $P_{\max}$, $\tau$, or $R_{\text{fmt}}$ magnitudes.
>
> We note that Eq.3 uses the scaled reward values. Each reward component is normalized to the range $[P_{\min}, P_{\max}]$, where $P_{\min}$ and $P_{\max}$ are selected based on the natural score range of the corresponding reward model.
>
> To obtain these bounds, we evaluate the reward model (Skywork-7B-RM in our work) on an equally distributed sample from the MMIF-23k dataset\footnote{\url{https://huggingface.co/datasets/ChrisDing1105/MMIF-23k}}. This procedure provides empirical estimates of $P_{\min}$ and $P_{\max}$, ensuring that all reward components lie on a comparable scale. As a result, no single term dominates the training due to magnitude differences.
>
> > [W2] Advantage normalization change: Removing std is motivated by difficulty-dependent bias, but there is no ablation on this specific choice within HARMO.
>
> We performed an ablation study using different normalization strategies, including REINFORCE++, RLOO, and GroupNorm (from DeepSeek-R1). We did not observe significant differences in overall performance across these strategies.
>
> However, the GRPO advantage function without the standard deviation consistently provided the best performance. Additionally, we observed that response lengths were more stable when the standard deviation was removed. This modification is also introduced and analytically justified in DR-GRPO (https://arxiv.org/abs/2503.20783)
>
> > [W3] The baselines in the experiments are not clearly expressed.
>
> We would like to clarify the baselines used in our experiments. For the 3B family, we compare our proposed HARMO model with Qwen2.5-VL-3B-Instruct, and for the 7B family, we compare with Qwen2.5-VL-7B-Instruct, ensuring a fair comparison within each model size.
>
> We would greatly appreciate it if the reviewer could provide additional details on which aspects of the baselines were unclear, so that we can address them thoroughly.
>
> > [W4] Open-ended scoring with cosine similarity to a single reference contradicts the claim that these tasks are not verifiable. It risks punishing diverse, equally good answers, especially in VQA with many acceptable phrasings.
>
> We agree that open-ended tasks are inherently more difficult to verify precisely compared to tasks such as math or coding. The use of cosine similarity to a single reference is only one variant of HARMO, as described in Table 1. To better handle diverse and out-of-distribution responses, we also conduct experiments using a reward model (RM) as the scorer within HARMO, which provides more flexible and robust evaluation for open-ended VQA tasks.
>
>
> > [Q1] How are $R^{\text{hybrid}}$, $R^{\lambda}$, and $R^{\text{fmt}}$ scaled or weighted relative to each other?}
> $R^{\text{hybrid}}$, $R^{\lambda}$, and $R^{\text{fmt}}$ are all scaled to the range $[P_{\min}, P_{\max}]$, specifically $[-10, 10]$.
>
> - $R^{\text{hybrid}}$ contains $R^{\text{RM}}$, which is a continuous score within $[P_{\min}, P_{\max}]$, and $R^{\text{rule}}$, which is scored as $-10$ or $10$ depending on the correctness of the example.
> -  $R^{\lambda}$ is treated as a penalty with a fixed score of $-10$.
> -  $R^{\text{fmt}}$ is scored as $-10$ or $10$ based on format correctness.
>
>
> > [Q2] How to differentiate which responses are verifiable and which are open-ended?
>
> We created a dataset in which each sample has a field answer\_format, classified as open-ended, numeric, or math_equation. Based on this information, we differentiate which responses are verifiable and which are open-ended during scoring.

---

### Official Review · Reviewer_qjJZ · 2025-10-26

**Soundness:** 3
**Presentation:** 3
**Contribution:** 2
**Rating:** 4
**Confidence:** 4

**Summary:**

This paper introduces HARMO, a novel Hybrid and Multi-Aspect Reward Modeling Optimization framework for aligning MLLMs. It tackles the limitations of monolithic rewards by integrating hybrid accuracy rewards (combining rule-based verification for verifiable tasks and model-based feedback for open-ended tasks) and multi-aspect behavioral rewards (including a generalized length penalty to prevent brevity-based reward hacking and a format adherence reward). Experiments show HARMO significantly improves MLLM performance, particularly in mathematical reasoning.

**Strengths:**

1. This paper introduces a novel hybrid reward system and a highly effective generalized dynamic length penalty, directly addressing critical limitations of current MLLM alignment methods.
2. This is a high-quality research with meticulous methodology, comprehensive ablation studies, robust empirical validation across diverse benchmarks, and statistical rigor.
3. This paper is well-written, logically structured, with clear explanations of complex concepts, and supported by informative figures, tables, and illustrative case studies.
4. This paper makes a substantial contribution by tackling fundamental MLLM alignment challenges, leading to significant performance improvements in reasoning, and offering a practical, scalable framework for future research.

**Weaknesses:**

1. Supplementary material images have low resolution, higher resolution is recommended.

**Questions:**

1. Given that the proposed hybrid and multi-aspect reward components (rule-based, model-based, length, format) and loss function are largely modality-agnostic and could be directly applied to pure-text LLM, how does HARMO's methodology offer a strong design specifically for cross-modal alignment that differentiates it from a generically applied LLM alignment technique?
2. While individual reward components like rule-based, model-based, length penalties (e.g., DAPO[1]), and format adherence (e.g., GRPO) have been previously explored, please clarify the specific novelty or unique differentiating factors of HARMO's combined reward formulation and its integration strategy that contribute to its claimed efficacy for MLLM alignment.
3. Your modification to the GRPO advantage function removes the standard deviation for "more stable and unbiased learning" based on an argument of "difficulty-dependent bias." Could you provide a more rigorous mathematical argument or empirical analysis to substantiate this claim, especially regarding the theoretical implications for policy gradient estimation and training stability?
4. The Skywork7B RM is central to your hybrid reward. Could you provide more details on its training data, particularly regarding its alignment with human preferences on diverse MLLM tasks? Are there known biases or limitations of this RM that could impact the HARMO training process?
[1] Yu Q, Zhang Z, Zhu R, et al. Dapo: An open-source llm reinforcement learning system at scale.

---

> ### Author Response · Authors · 2025-11-29
>
> We sincerely thank the reviewer for the detailed feedback. We appreciate the time and effort spent in evaluating our work and addressing both strengths and points for improvement. Our responses to each concern are provided below :
>
> > [W1] Supplementary material images have low resolution, higher resolution is recommended.
>
> Thanks for the note. We have updated the images in the revised version to higher resolution for better clarity.
>
> > [Q1] Given that the proposed hybrid and multi-aspect reward components (rule-based, model-based, length, format) and loss function are largely modality-agnostic and could be directly applied to pure-text LLM, how does HARMO's methodology offer a strong design specifically for cross-modal alignment that differentiates it from a generically applied LLM alignment technique?
>
> We acknowledge that the proposed hybrid and multi-aspect reward components (rule-based, model-based, length, format) are largely modality-agnostic and, in principle, could be applied to pure-text LLMs.
>
> Our core contribution lies in the systematic design and evaluation of the reward modeling methodology for cross-modal alignment. We focus our experiments on vision-language models (VLMs) because they present unique challenges compared to text-only LLMs, including limited preference data for RLHF, the need for reward models that support multimodality, and largely unexplored analytic and experimental evaluation. To address these challenges, we provide variants of the reward design (Table 2), such as the Embedding + Rule-based Hybrid Enhanced variant, which does not rely on preference data or an external reward model yet achieves competitive performance. Finally, our experimental study identifies the best-performing HARMO variant when using the Skywork-7B reward model in combination with multi-aspect rewards (Table~3)
>
> > [Q2] While individual reward components like rule-based, model-based, length penalties (e.g., DAPO[1]), and format adherence (e.g., GRPO) have been previously explored, please clarify the specific novelty or unique differentiating factors of HARMO's combined reward formulation and its integration strategy that contribute to its claimed efficacy for MLLM alignment.
>
> Our core contribution lies in reward modeling through a hybrid and multi-aspect approach, which differentiates HARMO from prior work:
> - Extension beyond deterministic verification: While GRPO (introduced in DeepSeek-R1-Zero) and DAPO focus on rule-based rewards for tasks with deterministic checks (e.g., math and code), HARMO combines rule-based and open-ended rewards within a unified hybrid formulation (Equation 1). This enables effective supervision for tasks that cannot be deterministically verified.
> - Introduction of multi-aspect rewards: HARMO incorporates length penalties, format rewards, and process-based feedback. We also analyze the individual contributions of each component through controlled ablation studies.
> - Quantitative analysis of reward effects: As shown in Tables 2 and 3, the improvements are attributable to our specific reward design, rather than the general concept of hybrid rewards.
>
> > [Q3] Your modification to the GRPO advantage function removes the standard deviation for "more stable and unbiased learning" based on an argument of "difficulty-dependent bias." Could you provide a more rigorous mathematical argument or empirical analysis to substantiate this claim, especially regarding the theoretical implications for policy gradient estimation and training stability?
>
> We performed an ablation study using different normalization strategies, including REINFORCE++, RLOO, and GroupNorm (from DeepSeek-R1). We did not observe significant differences in overall performance across these strategies.
>
> However, the GRPO advantage function without the standard deviation consistently provided the best performance. Additionally, we observed that response lengths were more stable when the standard deviation was removed. This modification is also introduced and analytically justified in DR-GRPO (https://arxiv.org/abs/2503.20783)
>
> > [Q4] The Skywork7B RM is central to your hybrid reward. Could you provide more details on its training data, particularly regarding its alignment with human preferences on diverse MLLM tasks? Are there known biases or limitations of this RM that could impact the HARMO training process? [1] Yu Q, Zhang Z, Zhu R, et al. Dapo: An open-source llm reinforcement learning system at scale.
>
> Skywork7B RM is an open-source model trained on a combination of human preference datasets and high-quality synthetic data; detailed information on its training data can be found in https://arxiv.org/pdf/2505.07263 . While any reward model may carry inherent biases from its training data, we do not expect these to significantly affect HARMO, as the method is designed to be agnostic to the choice of reward model and can incorporate alternative RMs if needed.

---

### Official Review · Reviewer_ipbD · 2025-10-31

**Soundness:** 3
**Presentation:** 3
**Contribution:** 3
**Rating:** 6
**Confidence:** 5

**Summary:**

The authors present a reward modeling framework, termed HARMO, for MLLMs alignment. The HARMO reward is a combination of rule-based, model-based, and behavioral rewards. Empirical evaluations on mathematical reasoning and VQA benchmarks demonstrate the effectiveness of the proposed reward model.

**Strengths:**

1. The idea of buidling a hybrid reward combining rule-based rewards and model-based rewards is insightful, as well as the finding of the strong bias towards shorter outputs for RL aligned models.

2. The hybrid reward yields significant performance gains over traditional RM-based baselines on a diverse set of benchmarks covering mathematical, general VQA, and OCR-based vision tasks.

**Weaknesses:**

1. It is not clear in Eq. (1) how to determine a response is verifiable and open-ended? When calcuating the cosine similarity, how to obtain the reference response? Then, a decision threshold is needed? Moreover, as I understand, determining whether a question-and-answer task is open-ended or verifiable primarily depends on the nature of the question, not the response.

2. In Eq. (3), why not use three weighting parameters to integrate the three reward components? How to guarantee a balance among them, giving the varying scales of their values?

3. To directly show the advantages of the HARMO reward model, I suggest that the authors can conduct experiments on some MM reward benchmarks, e.g., Multi-Modal Reward Bench (M. Yasunaga, arXiv:2502.14191v1, 2025), VL Reward Bench (L. Li, CVPR 2025) and MM-RLHF-Reward Bench (Y. Zhang, ICML, 2025).

4. The experimental results show that the HARMO significantly enhances math reasoning capability. Could you validate the method on more complex visual reasoning tasks in high-definition images? I suggest the author do some validation on the MME-RealWorld benchmark (Y. Zhang, ICLR 2025.)

**Questions:**

Please find the questions in the weaknesses part.

---

> ### Author Response · Authors · 2025-11-29
>
> We thank the reviewer for the thorough and insightful comments. We address each concern below.
>
> > [W1] It is not clear in Eq. (1) how to determine a response is verifiable and open-ended? When calcuating the cosine similarity, how to obtain the reference response? Then, a decision threshold is needed? Moreover, as I understand, determining whether a question-and-answer task is open-ended or verifiable primarily depends on the nature of the question, not the response.
>
> Each sample in our dataset contains the following fields:
>
> - message: a list of system/user messages, including image and text information.
> - reasoning: step-by-step ground truth reasoning leading to the answer.
> - answer: the ground truth final answer.
> - answer_format: the format of the answer (e.g., *open_ended*, *numeric*, *math_equation*).
>
> Based on the *answer_format* field, we differentiate whether a response is verifiable or open-ended during scoring. If helpful, we can also provide example data samples in the Appendix to further clarify this procedure.
>
> --------------------------------------------------------------------------------------------------------------------------------------------------------
>
> > [W2] In Eq. (3), why not use three weighting parameters to integrate the three reward components? How to guarantee a balance among them, giving the varying scales of their values?
>
> Thank you for the insightful question. Our ablation study on using weighting parameters for the three reward components did not show a significant impact on performance. Therefore, we conducted our main experiments using equal weighting for each component. To guarantee balance across components with different value ranges, we normalize each reward to a fixed scale $(P_{\min}, P_{\max})$, as shown in Eq.~(3). The complete normalization equations are provided in the Appendix.
>
> We set $P_{\min}$ and $P_{\max}$ to $-10$ and $10$, respectively. These values are obtained by evaluating the reward functions on an equally distributed MMIF-23k dataset (https://huggingface.co/datasets/ChrisDing1105/MMIF-23k).
>
> --------------------------------------------------------------------------------------------------------------------------------------------------------
>
> > [W3] To directly show the advantages of the HARMO reward model, I suggest that the authors can conduct experiments on some MM reward benchmarks, e.g., Multi-Modal Reward Bench (M. Yasunaga, arXiv:2502.14191v1, 2025), VL Reward Bench (L. Li, CVPR 2025) and MM-RLHF-Reward Bench (Y. Zhang, ICML, 2025).
>
> > [W4] The experimental results show that the HARMO significantly enhances math reasoning capability. Could you validate the method on more complex visual reasoning tasks in high-definition images? I suggest the author do some validation on the MME-RealWorld benchmark (Y. Zhang, ICLR 2025.)
>
> Thanks for the suggestion regarding additional evaluations. We plan to conduct experiments on the recommended MM reward benchmarks as well as the MME-RealWorld benchmark, and will include the results in the paper.

---

### Official Review · Reviewer_8ZQw · 2025-11-01

**Soundness:** 2
**Presentation:** 3
**Contribution:** 1
**Rating:** 2
**Confidence:** 4

**Summary:**

This paper proposes a hybrid reward optimization framework for aligning Multimodal Large Language Models (MLLMs). The framework combines model-based rewards, rule-based rewards to move beyond what the authors describe as "monolithic" single-signal rewards. The method is primarily evaluated on math and visual question-answering (VQA) tasks.

**Strengths:**

It's somewhat valuable to apply the rule-based + model-based rewards to a vision-language model on math/reasoning and VQA tasks and do the ablation study.

**Weaknesses:**

+ The primary issue with this paper is its lack of originality. The authors correctly identify that simple rule-based rewards are often limited to deterministic tasks and unsuitable for nuanced, open-ended feedback. However, the hybrid methodology they propose to solve this (combining different reward signals, including process-based feedback) is not new. A very similar paradigm was presented in the DeepSeek-R1 model, which was released substantially earlier (approx. 10 months ago) than this submission. The authors may mean DeepSeek-R1-**Zero** in line 52. Therefore, the core contribution of this paper appears to be an application of a pre-existing technique.
+ The paper claims to be about MLLM "alignment," which implies a broad improvement in helpfulness and safety across diverse tasks. However, the experiments and reward design are overwhelmingly focused on the math/reasoning domain.
+ The method fails to demonstrate a clear benefit. On the VQA tasks, the model's performance actually decreases slightly on docvqa and chartqa (7b) compared to the baseline. Also, this baseline is only an instruct-model (Qwen2.5-VL-instruct).

**Questions:**

The scores on lines 423 to 429 do not appear to match directly with those in Table 5. I guess the scores are relative improvements?

---

> ### Author Response · Authors · 2025-11-29
>
> We would like to thank the reviewer for the helpful and constructive comments. We address each weakness and question in the following.
>
>
> > [W1] The primary issue with this paper is its lack of originality. The authors correctly identify that simple rule-based rewards are often limited to deterministic tasks and unsuitable for nuanced, open-ended feedback. However, the hybrid methodology they propose to solve this (combining different reward signals, including process-based feedback) is not new. A very similar paradigm was presented in the DeepSeek-R1 model, which was released substantially earlier (approx. 10 months ago) than this submission. The authors may mean DeepSeek-R1-Zero in line 52. Therefore, the core contribution of this paper appears to be an application of a pre-existing technique.""
>
> Our core contribution is not limited to introduction of hybrid rewards itself, but rather the **systematic formulation, analysis, and evaluation** of reward components to improve accuracy across both verifiable (e.g., math) and non-verifiable (open-ended) tasks. Specifically:
>
> * Extension beyond deterministic verification:  While DeepSeek-R1-Zero primarily employs rule-based rewards for domains with deterministic checks (e.g., math and code), our work combines rule-based and open-ended rewards within a unified hybrid formulation (Equation 1). This enables effective supervision for tasks that cannot be deterministically verified.
> * Introduction of multi-aspect rewards: Our method incorporates length penalties, format rewards, and process-based feedback, and we analyze the individual contributions of each component in a controlled manner (Equation 3).
> * Quantitative analysis of reward effects: Table 2 and Table 3 show models demonstrating that the improvements arise from our specific reward design, rather than the general idea of using hybrid rewards.
>
> We have revised the version to explicitly refer DeepSeek-R1-Zero for better clarity and specificity.
>
> --------------------------------------------------------------------------------------------------------------------------------------------------------
>
> > [W2] The paper claims to be about MLLM "alignment," which implies a broad improvement in helpfulness and safety across diverse tasks. However, the experiments and reward design are overwhelmingly focused on the math/reasoning domain.
>
> We plan to extend our experiments to a broader set of tasks, including helpfulness and safety evaluations, and will provide these results to address this concern.
>
> --------------------------------------------------------------------------------------------------------------------------------------------------------
>
> > [W3] The method fails to demonstrate a clear benefit. On the VQA tasks, the model's performance actually decreases slightly on docvqa and chartqa (7b) compared to the baseline. Also, this baseline is only an instruct-model (Qwen2.5-VL-instruct).
>
> Table 3 shows that HARMO maintains performance on VQA benchmarks. While the 7B model exhibits a marginal decrease (less than 1 point) on DocVQA and ChartQA, this difference is very small, and we view it as within the typical variance observed across VQA evaluations.
>
> We hypothesis that additional high-quality data focused on VQA could further strengthen performance. Our current dataset composition (Table 1) consists of 71.7\% mathematical reasoning samples and 28.3\% VQA samples.
>
> --------------------------------------------------------------------------------------------------------------------------------------------------------
>
> > [Q1] The scores on lines 423 to 429 do not appear to match directly with those in Table 5. I guess the scores are relative improvements?
>
> Yes, the values in lines 423--429 correspond to relative improvements. In all tables, we report the \textbf{increment in absolute score}, computed as:
>
> Δ=HARMO−baseline
>
> The values on lines 423--429 additionally show the percentage improvement, calculated as:
>
> % Improvement = ((HARMO − baseline) / baseline) × 100
>
> The MATH evaluation reported is the aggregated score across the MathVerse-mini, MATH-Vision-test, and MathVista-mini benchmarks.

---

### Note · Authors · 2026-01-09

I have read and agree with the venue's withdrawal policy on behalf of myself and my co-authors.